# Recent Breakthroughs and Advancements in NO_x_ and SO_x_ Reduction Using Nanomaterials-Based Technologies: A State-of-the-Art Review

**DOI:** 10.3390/nano11123301

**Published:** 2021-12-06

**Authors:** Moazzam Ali, Ijaz Hussain, Irfan Mehmud, Muhammad Umair, Sukai Hu, Hafiz Muhammad Adeel Sharif

**Affiliations:** 1Centre of Excellence in Solid State Physics, University of the Punjab, Lahore 05422, Pakistan; mak.ssp@yahoo.com; 2Key Laboratory of the Ministry of Education for Advanced Catalysis Materials, Institute of Physical Chemistry, Zhejiang Normal University, Jinhua 321004, China; ijaz9292@gmail.com; 3Health Science Center, Shenzhen University, Shenzhen 518060, China; Irfan_Mehmud_1984@hotmail.com; 4College of Chemistry and Environmental Engineering, Shenzhen University, Shenzhen 518060, China; umair_uaf@hotmail.com (M.U.); 2110223076@email.szu.edu.cn (S.H.)

**Keywords:** NO_x_ and SO_x_ removal, wet-scrubbing, catalytic systems, Fe_3_O_4_ nanomaterial

## Abstract

Nitrogen and sulpher oxides (NO_x_, SO_x_) have become a global issue in recent years due to the fastest industrialization and urbanization. Numerous techniques are used to treat the harmful exhaust emissions, including dry, traditional wet and hybrid wet-scrubbing techniques. However, several difficulties, including high-energy requirement, limited scrubbing-liquid regeneration, formation of secondary pollutants and low efficiency, limit their industrial utilization. Regardless, the hybrid wet-scrubbing technology is gaining popularity due to low-costs, less-energy consumption and high-efficiency removal of air pollutants. The removal/reduction of NO_x_ and SO_x_ from the atmosphere has been the subject of several reviews in recent years. The goal of this review article is to help scientists grasp the fundamental ideas and requirements before using it commercially. This review paper emphasizes the use of green and electron-rich donors, new breakthroughs, reducing GHG emissions, and improved NO_x_ and SO_x_ removal catalytic systems, including selective/non-catalytic reduction (SCR/SNCR) and other techniques (functionalization by magnetic nanoparticles; NP, etc.,). It also explains that various wet-scrubbing techniques, synthesis of solid iron-oxide such as magnetic (Fe_3_O_4_) NP are receiving more interest from researchers due to the wide range of its application in numerous fields. In addition, EDTA coating on Fe_3_O_4_ NP is widely used due to its high stability over a wide pH range and solid catalytic systems. As a result, the Fe_3_O_4_@EDTA-Fe catalyst is projected to be an optimal catalyst in terms of stability, synergistic efficiency, and reusability. Finally, this review paper discusses the current of a heterogeneous catalytic system for environmental remedies and sustainable approaches.

## 1. Introduction

To keep life simple and comfortable isa concept as old as the beginning of time and this leads to the new inventions, discoveries and forward efforts. The quest for ease and a more convenient life strives to utilize the surrounding things and convert them into useful items. The start of the things always positive and man used the available items present in the surroundings. The same concept comes from burning or combustion (burning of substance by oxygen and generating heat i.e., energy accompanying flame i.e., light). This burning of fossil fuels has been observed from the earliest times and all people get advantages in different ways. Every new civilization improves it to meet their needs. Although the use of fossil fuels has created new customs, the overuse of combustion and its adverse effects cannot be denied [1,2]. Although, the energy obtained through combustion is very useful, it leaves adverse effect on marine life, ecosystem, human health and environment [3,4].

The rapid, cheap, and unlimited energy demands are the major reasons for producing electricity via combustion or fossil fuel. This results in an over demand for cheap energy and combustion of fossil-fuel eliminating toxic gases such as nitrogen oxides (NO_x_), sulfur dioxide/trioxide (SO_x_) and carbon monoxide (CO), which are rapidly increasing air-pollution [5]. The total contribution by fossil fuels for electricity production is shown in Figure 1 [6,7,8]. The basic reason is that fossil fuel combustion for electricity production is the major anthropogenic source of atmospheric pollutants. Therefore, coal is the most significant source of electrical energy, and it is projected to increase by a particular ratio each year for the next 25 years. Coal is cheap and readily available everywhere, and it is still believed that coal will be the considerable source for the next two decades for the production of power [9,10,11]. NO_x_ and SO_x_ containing flue gases were also released by the combustion of fuel and gas, but the concentration of these gases in coal-burning was roughly 2 to 3 times higher than that in gas burning [12]. Based on this calculation approximately 40 to 50% of the electrical energy of the world is being generated by coal-fired power plants [13]. Previously, several review studies were conducted to cover significant features of NO_x_ and SO_x_ elimination techniques, including wet scrubbing, SCR/SNCR etc. However, there is still a gap in the literature for a state-of-the-art review paper focused on a combination of using green and electron-rich donors, new breakthroughs, reducing GHG emissions, and improved NO_x_ and SO_x_ removal catalytic systems, including selective/non-catalytic reduction (SCR/SNCR) and other techniques (functionalization by magnetic nanoparticles; NP, etc.,) [14].

## 2. NO_x_ and SO_x_ Emission from Combustion

Industrialization and urbanization push the environment more adversely by producing more than 90% of NO_x_ because of fuel combustion. Furthermore, the pollutants from various sources such as unlimited combustion of industries, coal-fired power plants, on-road vehicles, aviation transport, gas and fuel refineries combustion exacerbate the situation daily. These sources left the worse effect on the atmospheric equilibrium by the emission of acidic gases (NO_x_, SO_x_) and particulate matter (PM), which are causing many negative impacts on environment, as well as for health [15,16]. The short overview of primary pollutants (NO_x_, SO_x_) is shown in Table 1 [17]. The infinite industrial exhaust in the lower atmosphere needs controlled for the sustainability of environmental balance by treating the toxic gases. Recently, various stringent laws and regulations have been developed worldwide to reduce the peril influence of toxic gases, especially by anthropogenic sources. The general trend for NO_x_ and SO_x_ emissions from power plants can be found such as in this Equation (1) [12].
Gas-emission > oil-emission > coal-emission(1)

### 2.1. Adverse Effect of NO_x_

The pollution is chronic and becomes acute after entering the human body. The most common and fundamental way in contact with air respiration and ingestion and skin contact also risk [18]. Even though the harmful effect of only NO_x_ is not irrevocable and thought to be toxic. This gas in the air and oxygen combination also generates many other secondary pollutants and forms photochemical smog [19,20]. For example, the NO is less harmful than NO_2_ and affects the eyes and throat. The more common and stable nitrogen oxides are given in Table 2 with their characteristics. As a result, the number of contaminants released into the atmosphere increases, and people are affected by blends and combinations of pollutants that may lead to more severe health issues [21].

### 2.2. NO_x_ Treatment Techniques

To meet the requirement of environmental protection organizations, a number of ways have been developed and executed on the individual industrial level for the treatment of NO_x_, SO_2_. Some techniques are used to decrease NO_x_ production during combustion, on the other hand, many ways are also applied to reduce the post-combustion. The overview of currently used abatement technologies is shown in Figure 2. The treatments include dry sorbent injection (DSI), selective noncatalytic reduction (SNCR), wet flue gas desulfurization (FGD), and selective catalytic reduction (SCR) [24,25]. For the treatment of NO_x_ and SO_2_, FGD and SCR are the two most commonly employed technologies.

#### 2.2.1. Selective Catalytic Reduction (SCR)

SCR is the most widely executed technique all over the world for flue gas and fuel combustion treatment on industrial and large-scale NO_x_ emission. This technique was introduced by Japan about 45 years ago and later also implemented in USA, Germany and many other European countries. In this technique, ammonia and air are mixed with NO gas in the presence of a catalyst (metallic-oxides), and under high temperature (250–600 °C), NO reduced into N_2_ and water ash are shown here [26,27].

Usually, the reduction efficiency of NO_x_ depends upon the addition of ammonia for a specific amount of NO_x_ (NO and NO_2_) as shown following, Equations (2)–(4).
4NH_3_ + 2NO + O_2_ → 2N_2_ + 6H_2_O(2)
4NH_3_ + 2NO_2_ + O_2_ → 3N_2_ + 6H_2_O(3)
2NO_2_ + NO + 4NH_3_ → 2N_2_ + 3H_2_O(4)

Because of its high acid strength (H^+^ ≤ −5.6) and remarkable Lewis and Brönsted acidity, Nb_2_O_5_ is the most extensively used catalyst of all [28]. The doping of Nb_2_O_5_ to Fe_2_O_3_ catalyst for the NH_3_-SCR reaction was initially described in 1985, and it is still in use today [29]. Although this phenomenon was discovered, the promotion mechanism was lost due to the limited characteristic procedures available at the time. According to a recent study, Ma et al. discovered that excellent SCR performance is achieved through using a Nb promoted CeZrO_x_ catalyst, which demonstrated high catalytic activity and N_2_ selectivity over a wide temperature range of 190–460 °C [30]. Mosrati et al. revealed that after Nb doping, a Ce/Ti catalyst displayed high activity and N_2_ selectivity, and Ce dispersion and strong acidity were essential factors in the catalyst’s high activity [31]. Nb addition can boost the catalyst’s anti-heavy metal poisoning activity as well as catalytic activity. Li et al. reported that modifying Mn/TiO_2_ catalysts with Nb increases Zn resistance [32]. Adding Nb to MnO_x_ (Mn-Nb mixed oxide catalysts) increases NO conversion and N_2_ selectivity due to increased Bronsted acidity and combining MnO_x_ and NbO_x_ [33]. In spite of the fact that the Nb modification boosted the acidity of the catalyst, it also reduced its reducibility, resulting in a synergistic relationship between acidity and reducibility in terms of the catalyst’s catalytic activity. In other words, the acidity and reducibility work together to enhance the catalytic activity [32].

#### 2.2.2. Selective Noncatalytic Reduction (SNCR)

It is possible to reduce NO by using a reagent, often ammonia or urea, at temperatures between 850 °C and 1175 °C using the SNCR approach [34]. This technique was also introduced by Japan in the late of 1970s [9,35]. In this experiment, the reagent, ammonia, combines with hydroxyl radicals (OH) to generate an amidogen radical (−NH_2_), which is as follows:NH_3_ + OH ⇌ NH_2_ + H_2_O(5)
when exposed to NO, this radical is selectively reactive and is most commonly involved in the following reactions:NH_2_ + NO ⇌ N_2_ + H_2_O(6)
NH_2_ + NO ⇌ NNH + HO(7)

The importance of reaction (8) can be attributed to the fact that it is a chain branching reaction that regenerates OH radicals required by the chain propagation reaction (5). However, a further reaction occurs with the NNH radical:NNH + NO ⇋ N_2_ + HNO(8)
NNH + M ⇋ H + NO + M(9)

More hydroxyl radicals are generated because of a chain reaction involving the H atom.

However, for normal operation of this technique the stoichiometric ration is about 4-times as compared to SCR for NO_x_ removal-reduction as shown in the following chemical reactions Equations (10) and (11) [36].
4CO(NH_3_)_2_ + NO → 2NH_3_ + CO_2_(10)
4NO + 4NH_3_ + O_2_ → 4N_2_ + H_2_O(11)

In stationary combustion systems, ammonium sulphate is being investigated as an addition for the simultaneous control of NO_x_ via SNCR) and deposition and corrosion and sulfation of alkali chlorides). According to Kristian et al., ammonium sulphate SNCR performance was assessed in a laboratory-scale flow reactor. NO reductions up to 95% were achieved at temperature range 1025–1075 °C, while using 5 and 10 w% solutions of aqueous ammonium sulphate, respectively, corresponding to ammonium sulphate/NO ratios over 1 [37]. According to reported literature, sulphur from ammonium sulphate is primarily emitted as SO_3_, even though SO_2_ is identified in high concentrations at very high temperature (e.g., over 1000 °C). There was evidence that adding KCl to the SNCR process promoted the reaction at lower temperatures, resulting in an additional 50 degrees Celsius of reduction potential. The high degree of KCl sulfation at or below 1000 °C was enabled by ammonium sulphate, suggesting the possibility of utilizing ammonium sulphate in full-scale combustion facilities to reduce NO_x_ and corrosion simultaneously. The experiments were analyzed in terms of a thorough kinetic model that was used. Even though the NO reduction at the optimum was significantly underestimated, the model accurately reproduced the SNCR experiments performed using ammonium sulphate. In addition, KCl sulfation was effectively documented; the enhancing effect of KCl on SNCR with ammonium sulphate was grossly overestimated. Possible explanations for this disparity were considered [22,38].

#### 2.2.3. Limitations of SCR and SNCR

However, the aforementioned methods are decent but still have limitations such as high temperature 700–1100 °C, less efficient for reduction-removal for SNCR, intrinsic use of additional chemicals, high cost, use of surplus freshwater, land area for FGD, and high temperature 300–750 °C, deactivation of catalysts and control of secondary pollutants such as ammonia (NH_3_) and hydrocarbons for SCR [39,40]. Another major drawback of this technology is that at high temperature for NH_3_ and NO gas mixture, the NH_3_ becomes oxidized at >400 °C. The oxidation of NH_3_ inhibits the further reduction or transformation of NO, which is not acceptable [41]. The universal law is that at high temperatures, the molecules of substances produce more energy, similarly, NH_3_ slips from the reaction chamber and escape into the atmosphere [35]. These parameters limit the applicability for execution on the industrial level. The low temperature and environmentally friendly techniques are highly appreciated.

#### 2.2.4. Common Solid Adsorbent Materials

Mostly post-combustion techniques are practiced due to the efficient reduction of NO_x_ and the objective of this work focuses on this method. Some other techniques also have been accessed, which are based on physical absorption of NO, including metal−organic frameworks (MOFs) such as (MIL: Materials of Institute Lavoisier) MIL-88A(Fe), MIL-96(Pt), MIL-100(Fe, Mn) and MIF-74(Co, Mn), activated carbon (AC), lime (CaSO_4_), zeolites and direct metal oxides. However, due to low efficiency, deactivation of materials, mist flue gas and selectivity either for NO or SO_2_ also inhibit the large-scale applications of these techniques [42,43]. Moreover, MOFs and zeolites are famous for their highest absorption capacity due to their large surface area and highest active sites for scavenging. Hence, the MOF’s materials are also extensively used for storage, transportation of gases and specially for NO storage, only MOFs were being chosen. To avoid the side effects (e.g., redox reaction of NO_x_), it is stored in special non-reacting cages made of MOFs and only released at the time of use [43,44].

MgO-organic component materials were synthesized using glucose and polyvinylpyrrolidone as raw materials in a one-step hydrothermal process. When combined with NO_x_ removal, the material is utilized to increase the efficiency of SO_2_ removal while simultaneously decreasing the competitive adsorption of both. It was determined if MgO-organic component/pure MgO/MgO (PVP modified)/MgO (glucose modified) improved the adsorption of SO_2_ and NO_x_ in the simulated coal-fired flue gas in the trials, and the comparison of the test findings was made. The MgO-organic components SO_2_ dynamic adsorption capacity was 0.3627 mmol/g, while NO_x_ was 0.2176 mmol/g, and the adsorption breakthrough time (time taken when the NO_x_ removal rate was 50%) was as long as 60 min (total flow rate of simulated flue gas is 200 mL/min, space velocity is 24,000 h^−1^, the reaction temperature is 100 °C, the concentration of SO_2_ and NO_x_ is 50% [45].

## 3. Low Temperature-Based Abatement Technique

### 3.1. Metal Ligand Absorption

Researchers and environmentalists have been working on alternative solutions to the concerns listed above and explored various approaches. Another critical thing to be discussed here is the NO solubility problem in water. Some other toxic gases such as CO_2_ and SO_2_ are easily treated via the alkali absorption method [46,47]. Since the solubility problem NO can be treated such as other gases, it required some combined or integrated system applied for NO reduction [48]. This solubility problem indicated the NO_x_ removal would be easier if chemical modifications were used for its absorption and reduction. Many ways came to notice, but most practice has been undertaken by wet scrubbing using Fe-EDTA. There are several reasons for considering flue gas treatment and reduction techniques, which are discussed herein. The MnO_x_/CNT_s_ catalysts were found to have unusual SCR activity at low temperatures when they were first synthesized. When using the optimal 1.2 percent MnO_x_/CNT_s_ catalyst at 80–180 °C, the NO conversion ranged between 57.4 and 89.2 percent. This occurred from the use of amorphous MnO_x_ catalysts, which have a higher ratio of Mn^4+^ to Mn^3+^ and O_S_ to (O_S_ + O_L_) than the crystalline MnO_x_ catalysts [49]. By impregnation and in situ deposition methods, the same Ce/Mn molar ratio was achieved in the preparation of Ce (1.0) Mn/TiO_2_ catalysts. In comparison to the impregnation-prepared Ce(1.0)Mn/TiO_2_-IP catalyst, the in situ deposition-prepared Ce(1.0)Mn/TiO_2_-SP catalyst demonstrated superior catalytic activity throughout a wide temperature range (150–300 °C) and at high-gas hourly-spaced velocities ranging from 10,500 to 27,000 h^−1^. Furthermore, the Ce(1.0)Mn/TiO_2_-SP catalyst produced by the in situ deposition approach has superior sulphur resistance to the Ce(1.0)Mn/TiO_2_-IP catalyst [50,51]. By using the citric acid–ethanol dispersion method, a variety of Gadolinium (Gd)-modified MnO_x_/ZSM-5 catalysts were produced and assessed using a low-temperature NH_3_-SCR reaction. Of them, the GdMn/Z-0.3 catalyst, which had a molar ratio of 0.3 for Gd to Mn, had the maximum catalytic activity, and it was capable of achieving a 100 percent NO conversion in the temperature range of 120–240 degrees Celsius. Furthermore, when tested in the presence of 100 ppm SO_2_, GdMn/Z-0.3 demonstrated superior SO_2_ resistance when compared to Mn/Z. It was demonstrated that such catalytic efficacy was primarily driven by surface chemisorbed oxygen species, a wide surface area, an abundance of Mn^4+^ and, a proper acidity and reducibility, and the of the catalyst, among other factors [52].

### 3.2. Metal Ligand Stability

Among transition metal chelating complexes, Fe-EDTA is the most favorable and stable chelate against the long-range of pH. Comparison for the metal chelate system are also shown in Figure 3. On top of that, ferrous (Fe(II)) has the great affinity towards the NO compared to other d-block elements. By taking advantage of this Fe-EDTA stability and greatest affinity, it has been used for NO_x_ removal from flue gas. Another advantage of this process (wet scrubbing) is the simultaneous removal of NO_x_ and SO_2_ under ambient conditions. Moreover, the NO_x_ interaction with Fe(II) is direct chemically binding and forms a very stable bond among other transition metals [12,53].

### 3.3. Principle of Gas Absorption

Basically, this concept is composed of two steps, (i) Fe-EDTA solution absorbs NO_x_ molecule via making metal-nitrosyl-complex and then (ii) this NO_x_ reduced into N_2_O, N_2_ and N-S compounds by utilizing the SO_2,_ which is a compulsory part of flue gas. This SO_2_ transformed into SO_3_^2−^ by alkali absorption and reduced NO_x_ by converting itself into SO_4_^2−^ ions. During the reduction of NO_x,_ the scrubber (Fe-EDTA) also regenerated irrespective the valence form of iron, Fe(II) or Fe(III). As Fe(III) is more stable due to more stability than Fe(II), it cannot be restored without the help of the electron donor externally. The number of electron donors used to reduce Fe(III) back to Fe(II) depends upon the source, condition and more important the NO_x_ reduction process.

## 4. Wet Scrubbing

### 4.1. Chemically Absorption by Metal-Ligand System

Several studies have demonstrated that wet scrubbing procedures are highly efficient, do not require intrinsic chemicals and are regenerable, requiring no additional fresh water or solution. This approach obtains EDTA-Fe(II) for chemical NO absorption by forming a metal-nitrosyl-complex, then, subsequently, reduced either by externally added sulphite solutions (SO_3_^2−^) or by transforming SO_2_ into SO_3_^2−^ ions through alkali absorption. The reductant may be activated carbon (AC) or an electrochemical system to provide electron for NO_x_ reduction and Fe-EDTA regeneration. As this reduction has been performed chemically so it is famous by name de-NO_x_/chem-de-NO_x_. Although wet scrubbing is good enough but due to oxygen contents which is a compulsory part of flue gas, make scrubber inactive by oxidation of EDTA-Fe(II) into EDTA-Fe(III) as shown in the followings Equations [54,55].
Fe(II)-[EDTA] + NO → [EDTA-Fe(II) (NO)](12)
SO_2_ + 2OH^−^ → SO_3_^2−^ + H_2_O(13)
Fe(II)-[EDTA(NO)] + SO_3_^2−^ + 2H^+^ + 2e^−^ → Fe(II)-[EDTA] + SO_4_^2−^ + N_2_ + H_2_O(14)
Fe(II)-[EDTA] + O_2_ + 4H^+^ → Fe(III)-[EDTA] + H_2_O(15)

### 4.2. NO_x_ Removal by Different Techniques

In the same way, there is also another analog technique of chemical wet scrubbing, i.e., bio-de-NO_x_. In this technique, the absorption step is similar to chem-de-NO_x,_ but some biological reduction methods used the transformation of NO_x_ regeneration of Fe-EDTA. The later step is carried out in the presence of microorganisms or bacteria under typical conditions; hence, it is called bio-de-NO_x_. The schematic diagram of bio-de-NO_x_ and chem-de-NO_x_ is shown in Figure 4. The difference in reduction or source of electron donor varies in both processes [56,57].

The absorption of NO_x_ for Fe-EDTA-based wet scrubbing is the same but reduction after making metal-nitrosyl-complex is different. The reduction in de-NO_x_/chem-de-NO_x_ is also carried out chemically by AC, sulphite solution; hence, it is known as chemical absorption and chemical reduction (CA-CR). In the same way, chemical absorption is integrated with the biological approach so that it is named a bio-de-NO_x_ system. In this integrated system, the reduction is carried out biologically via bacteria or microorganism by using glucose, ethanol, and chemical absorption and biological reduction (CA-BR). Both techniques CA-CR and CA-BR are accomplished at low temperature or mostly at room temperature, which is more suitable and cost-effective. As these processes are relatively similar, the problem (regeneration, efficiency, sustainability) of these technologies is also similar to some extent. Our newly published review paper on hybrid wet-scrubbing approaches for the removal of NO_x_ and SO_2_ can be referred to for further information [2,35].

## 5. Challenges of Wet Scrubbing Method

### 5.1. Regeneration of Fe-EDTA

The oxidized (EDTA-Fe(III)) fails to bind the NO from flue gas due to the stability of ferric (FeIII) ions [58]. Reducing Fe(III) into ferrous (FeII) can be accomplished in many ways but the preparation of reducing agents and constantly feeding-up reactors is not highly appreciated. To more sustainable and promising method, the system should be self-generated, i.e., it can be regenerate when used in operation, and the regeneration of Fe(II) is a pre-requisite of this technique. Therefore, the regeneration of Fe(II)-EDTA is not an easy task, so we have added some external reductants, which can assist the regeneration of Fe(II) from Fe(III) [59]. These externally added reductants help reduce the NO into N_2_ and regeneration Fe(II) after completing the cycle, as shown in Figure 5. Many researchers used a number of reductants to restore the Fe(II) back for NO absorption; some examples of reductants are also given in Table 3.

### 5.2. Secondary Pollutant

Although this system has many advantages such as it operates at low temperature, does not need extra intrinsic chemicals and scrubber solution is also regenerated, paradoxically, this system still has drawback by generating the secondary pollutant, nitrous oxide (N_2_O). This N_2_O is generated as an intermediate or by-products with N_2_ during NO_x_ reduction as shown in Figure 6 [55]. On the top of that the production of secondary pollutants, nitrous oxide (N_2_O) more water-soluble and separation from solution mixture is not easy. The main reason is the optimize pH for NO absorption and separation of N_2_O is different, hence, to grab the N_2_O almost impossible. The emission of this N_2_O is released by tail gas or treated gas, which is very secondary pollutant and very stable in the atmosphere.

## 6. Drawback of Wet Scrubbing

The absorption and reduction take place under the same conditions (specially pH) so that the recovery of N_2_O is not easy by this technique. Another critical problem is that the solubility of N_2_O in water is 5-fold than that of NO gas, so the separation of N_2_O is not easy [68]. It is one of the stable greenhouse gasses (GHG) and creates a number of problems after emission from NO_x_ treatment plant. N_2_O takes place by tail gas, which is a very serious issue because it is one of the GHG and causes serious problems such as ozone depletion substance (ODS) [69]. The accumulation of N_2_O in the atmosphere is again a critical problem that generates more N-oxides and, subsequently, causes atmospheric pollution.

## 7. Iron Oxide Nanoparticles

The retrieval of N_2_O is not possible from water, i.e., a significant amount is dissolved into water and finally released with tail gas, which again leads to the problem. This problem is not due to the solution mixture, it only relates to the applied condition and specially pH, and it is the main problem of this system. This problem can be resolved if the system operates under two different conditions i.e., absorption and reduction. This could be achieved easily if we transform the nature of Fe-EFTA into a solid form or by anchoring it into metallic particles solid particles. For metallic support, the iron-oxide nanoparticles (IONP) could be an excellent choice for many reasons. Iron is an indispensable part of chemistry, and a number of solid materials, composites and other catalytic tools are used on micro- and macro-levels. In addition, the overall reported iron-oxides are 16, but there are three main groups. Obviously, these three groups are due the stability and extensive occurring on the earth crust throughout the world. These iron oxides exist naturally and these groups with examples shown in Table 4 [70].

### 7.1. Synthesis of Magnetic Iron-Oxide Nanoparticles

Out of these oxides, ferric Fe(III) oxides exist abundantly due to oxygen and more stable electronic configuration. The Fe(III) oxides formed naturally into magnetite and green rust via Fe(II) conversion under specific conditions [70]. However, the magnetic compounds exist naturally and are synthesized in the laboratory. The synthesized magnetite (Fe_3_O_4_) are nano-sized super magnetic independent particles with different characteristics, structures, and its applications. There are a number of ways to synthesis these magnetic nanoparticles (NP) depending upon the characteristic application in nanoscience and nanotechnology [71,72]. Due to distinguished magnetic response, large surface area, and low cytotoxicity, the synthesis of Fe_3_O_4_ NP materials has been of hot interest in the last two decades. Therefore, several different ways have been discovered, to date, and a relative comparison of synthesis is shown in Figure 7 [73].

### 7.2. Solvothermal Synthesis

Out of these fabrication methods, the chemical method is attaining more interest and is mostly implemented due to high efficiency, low cost, simple easily adaptable and controllable [74]. Mostly chemical methods are achieved by basic iron salts, common acid-base and solvents used under easily achievable temperature and pH. The predominant techniques are different based on crystalline structure, size, and sometimes its magnetic properties [75]. The solvothermal method is more suitable for crystalline Fe_3_O_4_ NP than the oxygen-free environment, which is relatively expensive with controlled particles and shape [74]. The most common chemical reactions involved in this method during synthesis are given below Equations (16)–(21) [76].
FeCl_3_ + 3CH_3_COONa → Fe(CH_3_COO)_3_ + 3NaCl(16)
Fe(CH_3_COO)_3_ + 3H_2_O → 3CH_3_COOH + Fe(OH)_3_(17)
HOCH_2_CH_2_OH → HOCH_2_−CH=C=CH_2_ + 3H_2_O(18)
HOCH_2_CH_2_OH → CH≡CH + 2H_2_O(19)
Fe(OH)_3_ + HOCH_2_CH_2_OH → Fe(OH)_2_ + CH_2_CHO + 2H_2_O(20)
2Fe(OH)_3_ + Fe(OH)_2_ → Fe_3_O_4_ + 4H_2_O(21)

### 7.3. LaMer and Dinegar Model

The uniform dispersion of these synthesized particles is the priority for all applications and their subsequent results. The growth of monodispersed and uniform-sized metal-oxide (e.g., Fe-O) NP from their precursors depends upon the specific conditions, which remained without changing for a longer time. For more detailed understanding and justification, mostly two models explain this phenomenon and mechanism for completion of the growth of nuclei. The classic model, LaMer and Dinegar mechanism for growth into three stages, as shown in Figure 8, (I) the diffusion of monomers concentration gradually increases up to a specific supersaturation concentration essential for nucleation. (II) after supersaturation, a burst of nucleation occurs in which the solute diffuses from the solution for growth. (III) this growth proceeds by adding the monomer to the particle surface until the monodisperse final size particle is gained [77,78].

## 8. General Applications of Magnetite (Fe_3_O_4_)

Two kinds of iron oxides are magnetic in nature (i) magnetite (Fe_3_O_4_) and (ii) maghemite (Fe_2_O_3_, γ-Fe_2_O_3_). These magnetic particles are different regarding the composition of iron, oxygen and grown under different conditions. Both are magnetic and extensively used in various kinds of applications. In addition, magnetite magnetic nanoparticles (Fe_3_O_4_ NP) are receiving more interest from researchers due to the wide range of their application in numerous fields. The magnetic response is responsible for multifunctional applications in various sectors including energy storage, carbon capture, medical treatment, enzyme control, hydrogen storage, optical applications, nano electronics, and environmental remediation, as shown in Figure 9 [79,80,81]. These applications include utilization in the medical field such as for drug delivery, magnetic resonance imaging (MRI), and environmental treatments such as the removal of pollutants or contaminant particles [7]. Herein, Fe_3_O_4_ is employed for the treatment of flue gas due to its distinguished magnetic and stability characteristic [82,83].

Moreover, there are several other applications in which magnetic Fe_3_O_4_ NP is directly used, such as in loudspeakers, for damping and cooling agents, low friction seals, the active magnetic membrane used for biological reactor and microfluidic flow [84].

## 9. Functionalization of Magnetite NP

The predominant and outstanding characteristic of the Fe_3_O_4_ NP, which distinguished these NP over the other nanomaterials, is its magnetic property. Taking advantage of Fe_3_O_4_ NP, these are used in numerous fields by functionalization or coating of different materials with it [85,86]. The functionalization of Fe_3_O_4_ is very useful and makes for unique application in various fields and in the environment. This surface functionalization imparts characteristic properties, enhances stabilization, dispersion, interaction between the NP and target place, and stability over a long range of pH and temperature. The surface coating also avoids leaching, dissolution, aggregation, dispersion, surface charge exchange, increased surface area, porosity, and adsorption characterizations [75,87]. These unique characteristics make the material more functional, useful, and environmentally friendly.

### 9.1. Functionalized Magnetic NP for NO_x_ and SO_x_ Removal

A number of ways have been applied to make Fe_3_O_4_ useful for various fields, including drug and gene transportation, magnetic resonance imaging (MRI), filtration and purification, wastewater and atmospheric treatment purposes [74,82]. Additionally, the magnetism is distinguished property, making these particles more promising for the environment. Taking advantage of magnetically fast separation and easy recycling ability without a mechanical loss for several cycles made them more suitable for low operational cost and long-lasting adsorbents [74,88,89]. Recently, Fe(OH)_3_ and Fe_2_O_3_ catalysts for the selective catalytic reduction of NO_x_ with NH_3_ (NH_3_-SCR) were synthesized using a precipitation method and, subsequently, sulfated; the enhancing effect of SO_4_^2−^ functionalization on the performance of Fe_2_O_3_ catalyst in NH_3_-SCR was then examined. Results show that when compared to unmodified Fe_2_O_3_, the SO_4_^2−^-functionalized Fe_2_O_3_ catalysts had much higher SCR activity than non-treated Fe_2_O_3_ catalysts. In particular, the SO_4_^2−^/Fe(OH)_3_ catalyst demonstrates exceptional performance in NH_3_-SCR, with NO_x_ conversion rates of more than 80% at temperatures ranging from 250 to 450 °C; in addition, it exhibits good catalytic stability and resistance to H_2_O + SO_2_ in the presence of NH_3_. In addition, the functionalization of Fe_2_O_3_ NP by sulfuric acid inhibits its further growth and SO_4_^2−^ ions combined with Fe^3+^ to form a stable sulphate complex. Actually, it increases the active-sites and the acid strength, which can inhibit the ammonia over-oxidation on Fe_2_O_3_ and improve the NO_x_ performance of Fe_2_O_3_ [90,91]. According to another study, improving the low-temperature SO_4_^2−^-tolerant selective catalytic reduction (SCR) of NO_x_ with NH_3_ is an intractable problem due to the difficulty of decomposing accumulated sulphates below 300 °C [92,93]. Moreover, the wide range of surface modification on the magnetic support offers more specific and efficient catalysts. After surface modification, Fe_3_O_4_ is widely used for NO_x_, SO_x_ removal and reduction, while a few examples of these kinds of materials are enlisted in Table 5.

### 9.2. Surface Modification of NP by Ligands

The surface modification depends upon the stability of the loaded-layer and its application under specific conditions. The metal organic ligands (M-L) are relatively more stable after loading on Fe_3_O_4_ NP. A very common example of M-L is Ethylenediaminetetraacetic acid (EDTA) and Fe, extensively used for NO_x_ scrubbing treatment [103,104]. Although, this M-L system has been used for gas treatment for the last three decades. However, some very critical factors inhibit the industrial applications of EDTA-Fe for flue gas treatment. Those factors can be avoided if this EDTA-Fe load on the surface of Fe_3_O_4_ achieves this, then, the liquid system is transformed into a solid system. Moreover, previous studies also indicated the Fe_3_O_4_@EDTA system is very stable and showed high reusability [105,106]. The multi chelating ends (amino and carboxylic) of EDTA anchored the Fe_3_O_4_ solid NP and positively charged metal ions (FeIII/II) by improving the overall stability and catalytic properties and the proposed general structure is shown in Figure 10 [107,108]. The EDTA further provides stability to the developed system due to its high stability over long-range of pH and potential of making metal-complexes without reduction [109,110]. Hence, the Fe_3_O_4_@EDTA-Fe is expected to be an ideal catalyst regarding stability, synergistic efficiency, and reusability.

## 10. Conclusions

Overall, the removal of NO_x_ and SO_2_ from the atmosphere offers significant promise for commercialization on a wide scale for various applications. This study discovered that wet scrubbing is an effective approach that can be used in mild conditions and has a low operational cost due to the ease with which it can be set up. These are the main advantages of this technique, according to the findings. Nevertheless, the need for a substantial amount of freshwater and the need to regenerate the catalyst or scrubbing solution are the primary obstacles to its widespread deployment. To meet the requirements of environmental protection organizations, various methods for the treatment of NO_x_, SO_2_, and other pollutants have been devised and implemented at the individual industrial level. Some strategies are used to reduce NO_x_ production during combustion, while others reduce post-combustion NO_x_. Dry sorbent injection (DSI), selective noncatalytic reduction (SNCR), wet flue gas desulfurization (FGD), and selective catalytic reduction (SCR) are some of the treatment options available (SCR). FGD and SCR were determined to be the most often employed methods for treating NO_x_ and SO_2_, concurrently. N_2_O cannot be recovered using this wet scrubbing method since the absorption and reduction occur under the same conditions (in particular, pH). Another major issue is that the solubility of N_2_O in water is five times that of NO gas, making the separation of N_2_O a difficult operation. It is a stable greenhouse gas (GHG) that causes various difficulties when emitted by NO_x_ treatment plants. N_2_O is produced by tail gas, a severe concern because it is a greenhouse gas (GHG) that causes serious difficulties such as ozone depletion (ODS). To address these challenges, solvothermal production of solid magnetic nanomaterials such as iron-oxide nanoparticles and conducting functionalization of magnetite NP and surface modification of NP by ligands were used. It was also discovered that using various wet-scrubbing processes, the synthesis of solid iron-oxides such as magnetic (Fe_3_O_4_) NP is gaining popularity among researchers due to the vast range of applications in a variety of sectors. Furthermore, EDTA coatings on Fe_3_O_4_ NPs are commonly used because of their great stability over a wide pH range and their ability to form solid catalytic systems. Therefore, the Fe_3_O_4_@EDTA-Fe catalyst is expected to be the most stable, efficient, and reusable catalyst available in terms of stability, synergistic efficiency, and reusability. This review is beneficial for environmentalists, scientists, and specialists involved in minimizing the detrimental impacts of NO_x_ and SO_2_ at academic and commercial levels in the research and development sectors.

## Figures and Tables

**Figure 1 nanomaterials-11-03301-f001:**
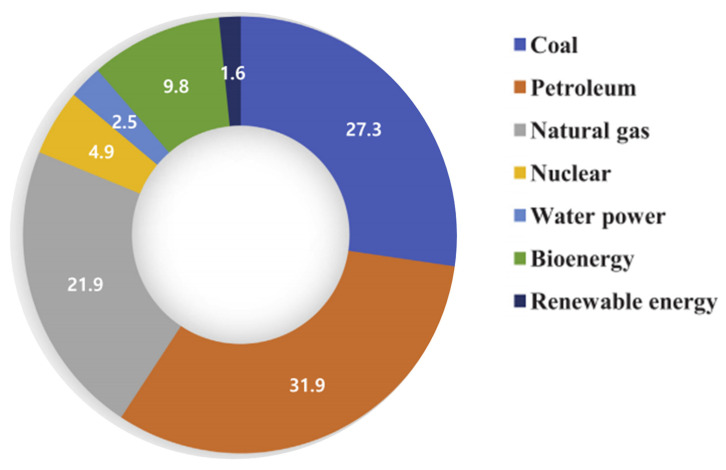
World demand forecast by primary energy source. Adapted with permission from [6]. Copyright, 2019 Elsevier.

**Figure 2 nanomaterials-11-03301-f002:**
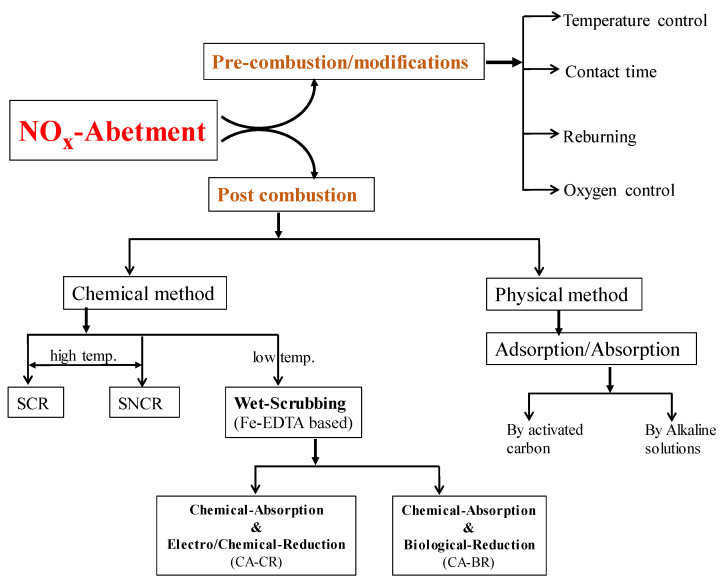
General overview of NO_x_ abatement systems by pre combustion and post combustion techniques, which further divided into specific methods based on operations.

**Figure 3 nanomaterials-11-03301-f003:**
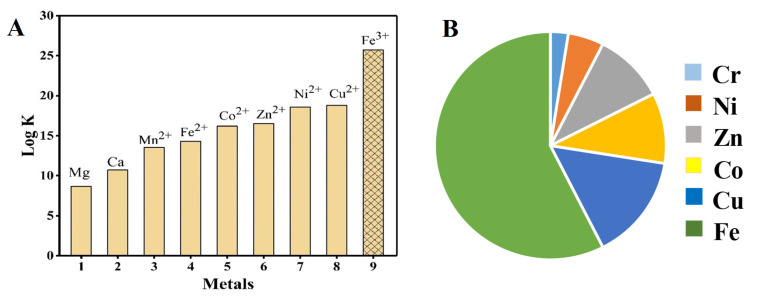
(**A**) Relative stability of Fe-EDTA with other metals, (**B**) relative NO gas and transition metal interactions.

**Figure 4 nanomaterials-11-03301-f004:**
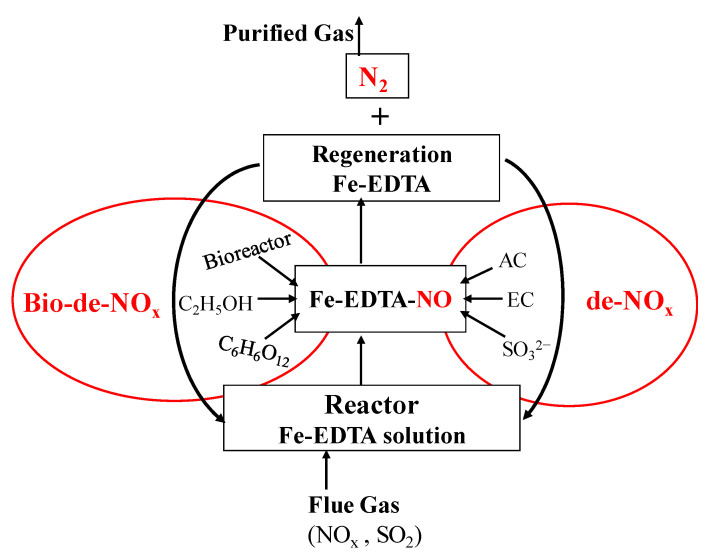
Schematic diagram for de-NO_x_ and bio-de-NO_x_.

**Figure 5 nanomaterials-11-03301-f005:**
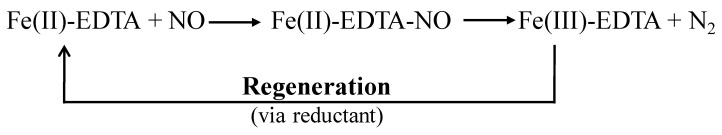
NO absorption, reduction and regeneration of Fe(II)-EDTA by external reductant.

**Figure 6 nanomaterials-11-03301-f006:**
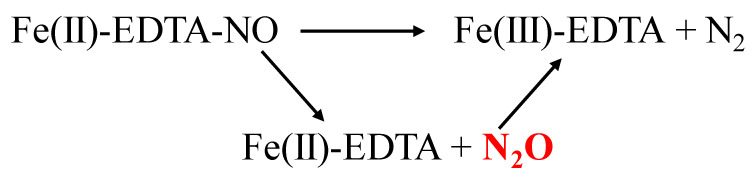
Reduction of NO via formation of N_2_O as an intermediate.

**Figure 7 nanomaterials-11-03301-f007:**
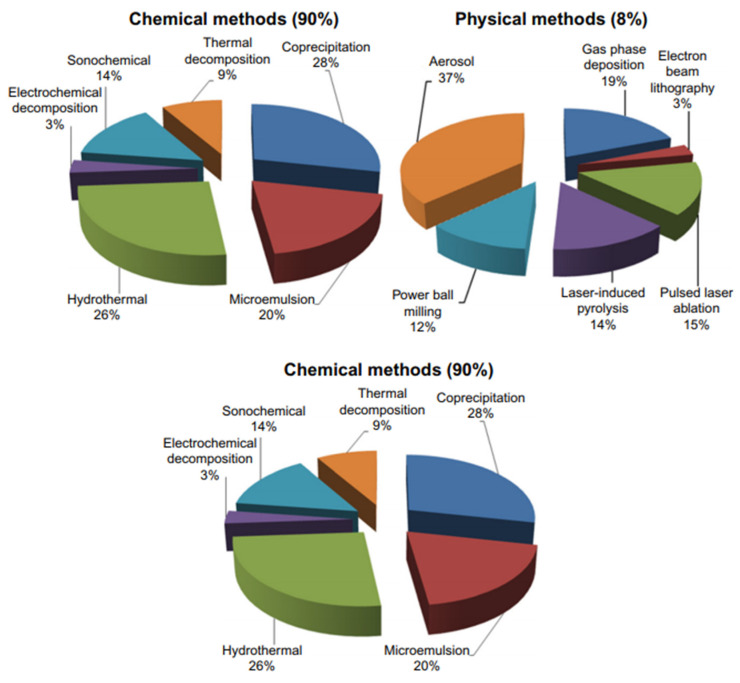
Relative comparison for the synthesis of iron oxide nanoparticles (IONP) by three different main categories that are further classified into different types based on sources and predominant principles. Adapted with permission from [73]. Copyright, 2016 Dovepress.

**Figure 8 nanomaterials-11-03301-f008:**
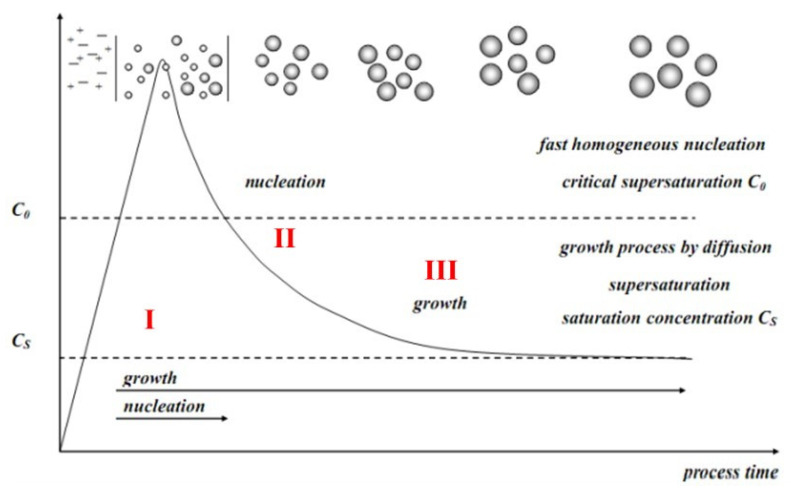
LaMer and Dinegar mechanism for monodisperse nanoparticles growth Adapted with permission from [78]. Copyright, 2016 American Chemical Society.

**Figure 9 nanomaterials-11-03301-f009:**
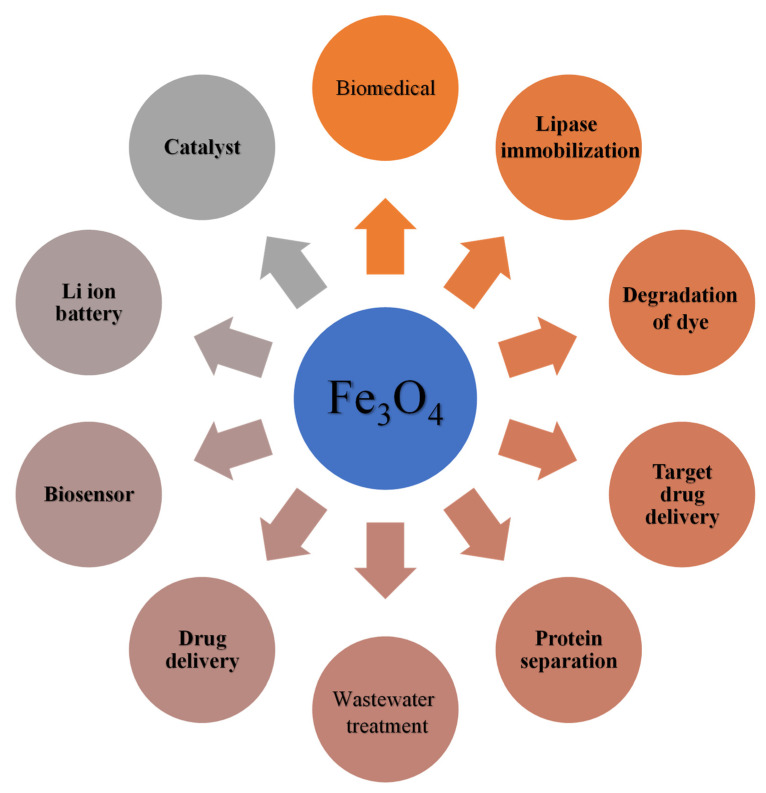
Applications of magnetite in various fields. Adapted with permission from [80]. Copyright, 2019 Springer.

**Figure 10 nanomaterials-11-03301-f010:**
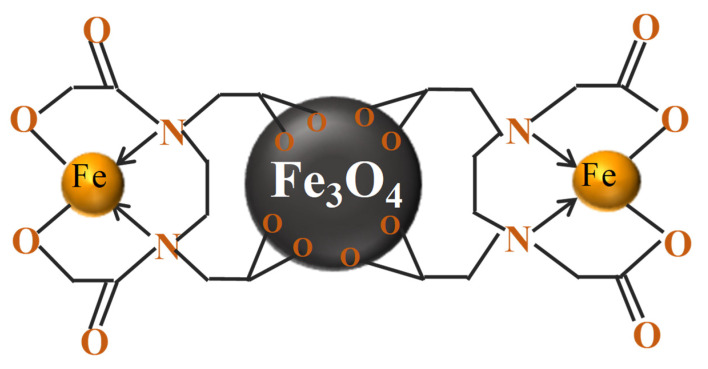
Proposed general structure of Fe_3_O_4_@EDTA-Fe.

**Table 1 nanomaterials-11-03301-t001:** Details of NO_x_ and SO_2_ combustion pollutants.

Name	Formula	Characteristic	Source	When and Where	Health Problems	Atmospheric Problems
Sulpher dioxide	SO_2_	Colour less	Coal plants, vehicles, industries	London 1952, Beijing, China 1985	Damages Lungs tissue, asthma	Smog, ozone formation, acid rain
Nitric oxide	NO	Colorless	Coal plants, vehicles, industries Combustion, nitric acid plants	Los Angeles 1940, Beijing, China 1985	Chronic respiratory, visibility, lungs damage, breast cancer, asthma attacks	Smog, ozone formation, acid rain, secondary pollutants
Nitrogen dioxides	NO_2_	Reddish-brown				

**Table 2 nanomaterials-11-03301-t002:** The most common existing nitrogen oxides and their physical properties of these oxides are given. Adapted with permission from [22]. Copyright, 2010 Elsevier.

Nitrogen Oxides	Color	Solubility (g·dm^−3^) [23]	State	Density (g·dm^−3^)
NO	Colorless	0.032	Gas	1.3402
N_2_O	Colorless	0.111	Gas	1.8
NO_2_	Red-brown	213.0	Gas	3.4
N_2_O_4_	Transparent	213.0	Liquid	1492.7 (273 K)
N_2_O_5_	White	500.0	Solid	20,508 K

**Table 3 nanomaterials-11-03301-t003:** The commonly used electron donor for regeneration of Fe-EDTA (ferric to ferrous) by different people during NO_x_ reduction via wet scrubbing (chemical reduction and biological reduction).

Chemical Reduction			Biological Reduction		
Reductants	Year	Ref.	Reductants	Year	Ref.
Na_2_SO_3_	1984	[60]	Ethanol	1999	[61]
SO_3_^2−^	1980	[62]	Acetate	2003	[63]
HSO_3_^2−^	1990	[55]	Glucose	2007	[64]
Polyphenolic compounds	1991	[59]	Gallic acid pyrogallol and tannic acid	1991	[65]
Hydrazine	1994	[1]			
Na_2_S_2_O_4_	2005	[66]			
Na_2_S	2006	[67]			

**Table 4 nanomaterials-11-03301-t004:** Most common and major groups of Iron-oxide based on valence.

Name	Oxidation State	Examples
Ferric oxides	Fe(III)	Ferrihydrite, goethite, lepidocrocite
Ferrous oxides	Fe(II)	Fe(II)O, Fe(II)(OH)_2_
Mixed-valent iron oxides	Fe(III) and Fe(II)	Magnetite, green rust (GR)

**Table 5 nanomaterials-11-03301-t005:** Functionalization of Fe_3_O_4_ by different organic/inorganic layers used for NO_x_ removal and NO_x_ reduction under different systems.

System/Material	Purpose	Reductant/Name of Technique	Year	Ref.
Fe_3_O_4_-Chitosan	NO_x_ removal	C_6_H_12_O_6_, microorganisms (Biological reduction)	2012	[94]
Fe_3_O_4_-poly(styrene–glycidylmethacrylate)	NO_x_ removal	By microorganisms (Biological reduction)	2013	[95]
Fe_3_O_4_	De-NO_x_	SCR	2010	[96]
Fe_3_O_4_/rGO	NO_x_-sensing	Injection of NO in reacting chamber	2016	[97]
Fe_3_O_4_/*mpg*-C_3_N_4_	NO_x_ oxidation and storage	Photocatalytic oxidation under visible light	2019	[98]
Fe_3_O_4_-TiO_2_	NO_x_ removal	Adsorption fixed-bed reactor and High temperature	2016	[99]
Fe_3_O_4_ and Fe_2_O_3_	De-NO_x_	SCR by NH_3_	2009	[100]
Rod-Shaped Fe_2_O_3_	NO_x_ reduction	SCR by NH_3_	2012	[101]
Fe_3_O_4_@CuS	Hg capture (Flue gas treatment)	Adsorption fixed-bed reactor	2018	[102]

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
