# Peer review of "Recent Breakthroughs and Advancements in NOx and SOx Reduction Using Nanomaterials-Based Technologies: A State-of-the-Art Review"

_nanomaterials, 2021, doi:10.3390/nano11123301_

Round 1

Reviewer 1 Report

Comments to the Paper titled: Recent Breakthroughs and Advancements in NOx and SO2 Reduction Using Nanomaterials-based Technologies: A state-of- the-art review.

In general a very poor English language has been used though the text of the Paper.  Some sentences lack the verb therefore it is difficult to understand the intended meaning. Most sentences use inappropriate language for a scientific paper or unclear (several examples are reported below) and need to be rephrased. In general the poor language prevent the reader understanding the meaning of the sentences.

A full language revision is necessary before publication.

Lines 44-45: put forward the life on 44 the new customs (inappropriate language for a scientific paper)

Lines 46-47: If this combustion is making life easy (inappropriate language for a scientific paper)

Lines 57-59 : As the fuel and gas combustion also released NOx and SO2 containing flue gas but the concentration of these gas about 2 to 3 folds in coal burning (unclear).

Lines 75-78: The general trend for NOx and SO2 emissions from power plants can be found like this eq. (1).   Gas>oil>coal (inappropriate language for a scientific paper).

Lines 88-90: Therefore, the number of contaminants emitted to the air, and we get affect by blend and combined pollutants which may lead towards more serious disease. (unclear)

Lines 107-108: Out of these FGD and SCR gener ally used for treatment of NOx and SO2 simultaneously (needs to be rephrased).

Lines 212-213: To resolve the above problems people start working on some other ways and tried 212 many things (unacceptable language for a scientific paper).

Author Response

Comments to the Paper titled: Recent Breakthroughs and Advancements in NOx and SO2 Reduction Using Nanomaterials-based Technologies: A state-of- the-art review.

In general a very poor English language has been used though the text of the Paper.  Some sentences lack the verb therefore it is difficult to understand the intended meaning. Most sentences use inappropriate language for a scientific paper or unclear (several examples are reported below) and need to be rephrased. In general the poor language prevent the reader understanding the meaning of the sentences.

Response: We would like to thank the reviewer for constructive comments and appreciate his/her detailed evaluation and critical comments to bring homogeneity in describing our product. We have revised the manuscript as per recommendations and believe that valuable comments by the respected reviewer helped us to improve the quality of our article.

  1. A full language revision is necessary before publication.

Response: We would like to express our gratitude to the reviewer for his or her insightful comments. We have enhanced the manuscript's English in the amended version.

  1. Lines 44-45: put forward the life on 44 the new customs (inappropriate language for a scientific paper)

Response: We have enhanced the manuscript's English in the amended version.

“Every new culture improves it to meet their needs. Although the use of fossil fuels has created new customs, the overuse of combustion and its negative effects cannot be denied”.

  1. Lines 46-47: If this combustion is making life easy (inappropriate language for a scientific paper)

Response: Thank you for valuable suggestion, we have revised the sentence as follows:

“In order to make living easier, the combustion of fossil fuels has also had a negative impact on the maritime environment”.

  1. Lines 57-59: As the fuel and gas combustion also released NOx and SO2 containing flue gas but the concentration of these gases about 2 to 3 folds in coal burning (unclear).

Response: Thank you for valuable suggestions and we have revised the mentioned sentence and given below for your reference:  

“NOx and SO2 containing flue gases were also released by the combustion of fuel and gas, but the concentration of these gases in coal burning was roughly 2 to 3 times higher than that in gas burning” 

  1. 5. Lines 75-78: The general trend for NOx and SO2 emissions from power plants can be found like this eq. (1).   Gas>oil>coal (inappropriate language for a scientific paper).

Response: Thank you for valuable suggestion, we have revised the sentence as follows:

For NOx and SO2 emissions from power plants, the following equation can be used to determine the overall trend: (1).

Gas-emission>oil-emission>coal-emission  

  1. Lines 88-90: Therefore, the number of contaminants emitted to the air, and we get affect by blend and combined pollutants which may lead towards more serious disease. (unclear)

Response: Thank you for valuable suggestion, we have revised the sentence as follows:

“As a result, the number of contaminants released into the atmosphere increases, and people are affected by blends and combinations of pollutants that may lead to more serious health issues”

  1. Lines 107-108: Out of these FGD and SCR gener ally used for treatment of NOx and SO2 simultaneously (needs to be rephrased).

Response: Thank you for valuable suggestion, we have revised the sentence as follows:

“For the treatment of NOx and SO2, FGD and SCR are the two most commonly employed technologies”

  1. Lines 212-213: To resolve the above problems people start working on some other ways and tried 212 many things (unacceptable language for a scientific paper).

Response: Thank you for valuable suggestion, we have revised the sentence as follows:

“Scientists and environmentalists have been working on alternative solutions to the concerns listed above and explored a variety of approaches”.  

Reviewer 2 Report

This manuscript aimed at reviewing the recent breakthroughs and advancements in NOx and SOx reduction using nanomaterials. The topic is interesting, which have been reviewed by many groups. However, as a reviewer I have few concerns which need to be addressed prior to reaching any decision. My comments are as follows

What is the novelty and value of this review? What are the motivations of this study? What new or valuable information can be provided by this review?

Many figures and tables were provided in the manuscript, This manuscript lacks own creative figures and tables, all figures and tables are adopted with or without permission???.  while most of them I think are unnecessary. As we know, a good review paper should focus on the analyses and discussion according to the recent advances, rather than just a simple summary or even repeat of this work.

Title: “A state-of-the-art review” this word is not justified, why NOx and only SO2, not SOx, there is a confusion throughout the manuscript.

Introduction: provide up to date literature review and in the end explain how the paper differs from others, State specific objectives.

Check your heading and subheading numbering

NOx and SO2 emission from combustion: better to focus and cover SOx too,

Section 2.2. NOx treatment techniques: better to have summary of data in own creative table form

Section 3. Low temperature-based abatement technique and section 4. Wet scrubbing: not well covered as per title

Section 7. Solid magnetic nanoparticles and Section 8. General applications of magnetite (Fe3O4): not well covered as per title

Section  9. Functionalization of magnetite NP: cover SOx too

State main findings in the conclusions

Author Response

This manuscript aimed at reviewing the recent breakthroughs and advancements in NOx and SOx reduction using nanomaterials. The topic is interesting, which have been reviewed by many groups. However, as a reviewer I have few concerns which need to be addressed prior to reaching any decision. My comments are as follows

Response: We would like to thank the reviewer(s) for constructive comments and recommending our work for publication. We have revised the manuscript according to the suggestions and believe that valuable comments of the respected reviewer(s) helped us to improve the quality of our article. 

  1. What is the novelty and value of this review? What are the motivations of this study? What new or valuable information can be provided by this review?

Response: Thank you for your comment.

Previously, a number of review studies were conducted to cover significant features of NOX and SOX elimination techniques. However, it is still gap in a state-of-the-art review paper in combination of using green and electron-rich donors, new breakthroughs, reducing GHG emissions, and improved NOx and SOx removal catalytic-systems, including selective/non-catalytic reduction (SCR/SNCR) and other techniques (functionalization by magnetic nanoparticles; NP etc.,)

  1. Many figures and tables were provided in the manuscript, This manuscript lacks own creative figures and tables, all figures and tables are adopted with or without permission???.  while most of them I think are unnecessary. As we know, a good review paper should focus on the analyses and discussion according to the recent advances, rather than just a simple summary or even repeat of this work.

Response: Thank you for taking the time to comment. Each figure was used with permission, and we have already submitted all the permission letters to the editor of "Nanomaterials" with the information.

  1. Title: “A state-of-the-art review” this word is not justified, why NOx and only SO2, not SOx, there is a confusion throughout the manuscript.

Response: Thank you for your feedback; we apologize for any confusion; we have amended the title as follows.

Recent Breakthroughs and Advancements in NOx and SOx Reduction Using Nanomaterials-based Technologies: A state-of-the-art review

  1. Introduction: provide up to date literature review and in the end explain how the paper differs from others, State specific objectives.

Response: Thank you for your important recommendation, we have made some further additions to the amended version, which are as follows:

Previously, a number of review studies were conducted to cover significant features of NOX and SOX elimination techniques, including wet scrubbing, SCR/SNCR etc. However, it is still gap in a state-of-the-art review paper in combination of using green and electron-rich donors, new breakthroughs, reducing GHG emissions, and improved NOx and SOx removal catalytic-systems, including selective/non-catalytic reduction (SCR/SNCR) and other techniques (functionalization by magnetic nanoparticles; NP etc.,)

  1. Check your heading and subheading numbering

Response: Thank you for your suggestion and remark. We double-checked the amended manuscript's number of headings and subheadings.

  1. NOx and SO2emission from combustion: better to focus and cover SOx too.

Response: Thank you for your suggestion and remark. Included the SO2 studies as well in the section of 2.3.4. of the revised manuscript.

MgO-organic component materials were synthesised using glucose and polyvinylpyrrolidone as raw materials in a one-step hydrothermal process. When combined with NOx removal, the material is utilised to increase the efficiency of SO2 removal while simultaneously decreasing the competitive adsorption of both. It was determined if MgO-organic component/pure MgO/MgO (PVP modified)/MgO (glucose modified) improved the adsorption of SO2 and NOx in simulated coal-fired flue gas in the trials, and the comparison of the test findings was made. The MgO-organic component's SO2 dynamic adsorption capacity was 0.3627 mmol/g, while NOx was 0.2176 mmol/g, and the adsorption breakthrough time (time taken when the NOx removal rate was 50%) was as long as 60 min (total flow rate of simulated flue gas is 200 ml/min, space velocity is 24000 h-1, reaction temperature is 100 °C, concentration of SO2 and NOx is 50%

  1. Section 2.2. NOx treatment techniques: better to have summary of data in own creative table form

Response: Thank you for your suggestion and critical remark. Figure 2 summarizes Section 2.2, which we believe is more complete comprehensive than a table. We have also obtained permission to use Figure 2 in this publication.

  1. Section 3. Low temperature-based abatement technique and section 4. Wet scrubbing: not well covered as per title

Response: We appreciate the input and constructive criticism of the reviewer. Further details about low-temperature abatement techniques have been provided to in the section 3.

The MnOx/CNTs catalysts were found to have exceptional SCR activity at low temperatures when they were first synthesised. When using the optimal 1.2 percent MnOx/CNTs catalyst at 80-180 °C, the NO conversion ranged between 57.4 and 89.2 percent. This occurred from the use of amorphous MnOx catalysts, which have a higher ratio of Mn4+ to Mn3+ and OS to (OS+OL) than the crystalline MnOx catalysts. [https://doi.org/10.2174/1573413716999200812130206]. By impregnation and in situ deposition methods, the same Ce/Mn molar ratio was achieved in the preparation of Ce (1.0) Mn/TiO2 catalysts. In comparison to the impregnation-prepared Ce(1.0)Mn/TiO2-IP catalyst, the in-situ deposition-prepared Ce(1.0)Mn/TiO2-SP catalyst demonstrated superior catalytic activity throughout a wide temperature range (150–300 °C) and at high gas hourly space velocities ranging from 10,500 to 27,000 h-1. Furthermore, the Ce(1.0)Mn/TiO2-SP catalyst produced by the in situ deposition approach has superior sulphur resistance to the Ce(1.0)Mn/TiO2-IP catalyst [https://doi.org/10.1016/j.ces.2021.116588]. By using the citric acid–ethanol dispersion method, a variety of Gadolinium (Gd)-modified MnOx/ZSM-5 catalysts were produced and assessed using a low-temperature NH3-SCR reaction. Of them, the GdMn/Z-0.3 catalyst, which had a molar ratio of 0.3 for Gd to Mn, had the maximum catalytic activity, and it was capable of achieving a 100 percent NO conversion in the temperature range of 120–240 degrees Celsius. Furthermore, when tested in the presence of 100 ppm SO2, GdMn/Z-0.3 demonstrated superior SO2 resistance when compared to Mn/Z. It was demonstrated that such catalytic efficacy was primarily driven by surface chemisorbed oxygen species, a wide surface area, an abundance of Mn4+ and, a proper acidity and reducibility, and the of the catalyst, among other factors. [https://doi.org/10.3390/catal11030324].

While our newly published review paper on hybrid wet scrubbing approaches for the removal of NOx and SO2 can be referred to for further information [https://doi.org/10.1016/j.chemosphere.2021.129695].

  1. Section 7. Solid magnetic nanoparticles and Section 8. General applications of magnetite (Fe3O4): not well covered as per title

Response: Thank you for your critical comments, we apologize for the mistake; we have revised the heading of the section 7 and 8 as follows:

“7. Iron oxide nanoparticles”. “8. General applications of iron oxide nanoparticles (Fe3O4)”

  1. Section  9. Functionalization of magnetite NP: cover SOx too.

Response: Thank you for your suggestion and remark. We have added SOx studies in section 9 of the revised manuscript as follows:

Recently, Fe(OH)3 and Fe2O3 catalysts for the selective catalytic reduction of NOx with NH3 (NH3-SCR) were synthesised using a precipitation method and subsequently sulfated; the enhancing effect of SO2-4 functionalization on the performance of Fe2O3 catalyst in NH3-SCR was then examined. Results show that, when compared to unmodified Fe2O3, the SO42-functionalized Fe2O3 catalysts had much higher SCR activity than unmodified Fe2O3 catalysts. In particular, the SO42-/Fe(OH)3 catalyst demonstrates exceptional performance in NH3-SCR, with NOx conversion rates of more than 80% at temperatures ranging from 250 to 450°C; in addition, it exhibits good catalytic stability and resistance to H2O+SO2 in the presence of NH3. In conclusion, it was discovered that functionalization with sulfuric acid can inhibit the growth of Fe2O3 grains; furthermore, SO42- combines with Fe3+ to form the sulphate complex, resulting in an increase in the number of surface acid sites and the acid strength, which can inhibit the ammonia over-oxidation on Fe2O3 and improve the NOx performance of Fe2O3 [https://doi.org/10.1016/S1872-5813(20)30025-6]. According to another study, improving the low-temperature SO2-tolerant selective catalytic reduction (SCR) of NOx with NH3 is an intractable problem due to the difficulty of decomposing accumulated sulphates below 300°C. [https://doi.org/10.1021/acs.est.9b00435]  

  1. State main findings in the conclusions

Response: Thank you for your suggestion and remark. We have added the main findings in the conclusion of the revised manuscript as follows:

“It was also discovered that, using various wet-scrubbing processes, the synthesis of solid iron-oxides such as magnetic (Fe3O4) NP is gaining popularity among researchers due to the vast range of applications it has in a variety of sectors. Furthermore, EDTA coatings on Fe3O4 NPs are commonly used because of their great stability over a wide pH range and their ability to form solid catalytic systems. Therefore, the Fe3O4@EDTA-Fe catalyst is expected to be the most stable, efficient, and reusable catalyst available in terms of stability, synergistic efficiency, and reusability”.  

Round 2

Reviewer 2 Report

The authors have made sufficient modifications, and I suggest that this paper be accepted but this review lack of own created tables based on research quantitative data , there are some tables (1 to 4) but these tables are not required in this review article, these information available in the books

Author Response

Thank you for comment and suggestion

We agree with the reviewer's points; nevertheless, we used different tables in varied combinations to make the information of our paper more accessible to general readership. In these tables, we made use of proper references.